# TRP Channels in Cancer: Signaling Mechanisms and Translational Approaches

**DOI:** 10.3390/biom13101557

**Published:** 2023-10-22

**Authors:** Matilde Marini, Mustafa Titiz, Daniel Souza Monteiro de Araújo, Pierangelo Geppetti, Romina Nassini, Francesco De Logu

**Affiliations:** Department of Health Sciences, Clinical Pharmacology and Oncology Section, University of Florence, 50139 Florence, Italy; matilde.marini@unifi.it (M.M.); mustafa.titiz@unifi.it (M.T.); daniel.souzamonteirodearaujo@unifi.it (D.S.M.d.A.); pierangelo.geppetti@unifi.it (P.G.); francesco.delogu@unifi.it (F.D.L.)

**Keywords:** ion channel, transient receptor potential (TRP) channels, calcium, tumor proliferation, tumor metastasis

## Abstract

Ion channels play a crucial role in a wide range of biological processes, including cell cycle regulation and cancer progression. In particular, the transient receptor potential (TRP) family of channels has emerged as a promising therapeutic target due to its involvement in several stages of cancer development and dissemination. TRP channels are expressed in a large variety of cells and tissues, and by increasing cation intracellular concentration, they monitor mechanical, thermal, and chemical stimuli under physiological and pathological conditions. Some members of the TRP superfamily, namely vanilloid (TRPV), canonical (TRPC), melastatin (TRPM), and ankyrin (TRPA), have been investigated in different types of cancer, including breast, prostate, lung, and colorectal cancer. TRP channels are involved in processes such as cell proliferation, migration, invasion, angiogenesis, and drug resistance, all related to cancer progression. Some TRP channels have been mechanistically associated with the signaling of cancer pain. Understanding the cellular and molecular mechanisms by which TRP channels influence cancer provides new opportunities for the development of targeted therapeutic strategies. Selective inhibitors of TRP channels are under initial scrutiny in experimental animals as potential anti-cancer agents. In-depth knowledge of these channels and their regulatory mechanisms may lead to new therapeutic strategies for cancer treatment, providing new perspectives for the development of effective targeted therapies.

## 1. Introduction

According to the World Health Organization (WHO), cancer comprises a large group of diseases characterized by the rapid growth of abnormal cells that invade neighboring parts of the body, with the capacity to spread to other organs. More than 100 types of cancer have been identified so far [1]. Cancer represents a leading cause of death worldwide, causing nearly 10 million deaths in 2020 [2]. The leading cause in the development of cancer is an abnormal proliferation of cancer cells, which, rather than responding appropriately to the signals that regulate normal cell behavior, grow and divide in an uncontrolled manner. However, because of the large variability in cancers, tissues, organs of origin, predisposing or causal agents, genetic influence, and differential response to pharmacological treatments, the identification of common causes, mechanisms, and treatments is practically impossible. Notwithstanding, an improved knowledge of specific transduction signaling pathways may offer novel possibilities for innovative targeted treatments.

Ion channels are integral membrane proteins containing an aqueous pore that facilitates the mobilization of certain ions between cell compartments playing an essential role in cell functioning [3]. They regulate different cellular pathways, including cell proliferation, migration, apoptosis, and differentiation, to maintain normal tissue homeostasis. During the phenotypic changes that lead from a normal epithelial towards a cancer cell, a series of genetic/epigenetic changes, among other functions, may affect the activity of the ion channels [4]. Ion transport across the cell membrane has a crucial role in fundamental tumor cell functions [5], such as cell migration and cycle progression [6], cell volume regulation, proliferation, and death [7,8], which play critical roles in tumor cell survival and metastasis [9].

Increasing awareness of the ion channel’s role in tumor progression has led to the consideration of cancer as a channelopathy, or as a disease characterized by a profound alteration in ion channel function [10]. A special family of ion channels, called mechano-gated ion channels, includes the prototypical mechanosensitive piezo channels, which respond to mechanical stimuli such as changes in membrane tension or force [11]. Another family of ion channels, the transient receptor potential (TRP) channels, are opened by physical (mechanical and thermal) and chemical stimuli [12]. The great sensitivity of mechanosensitive ion channels to modifications in matrix stiffness is another significant feature [13]. Mechanical signals that operate through mechanosensitive ion channels during tumor growth have an impact on the microenvironment as well as cancer cells [14]. It has been reported that there is an association between gliomas and piezo channels in the regulation of tissue stiffness and tumor mitosis. Piezo1 sustains focal adhesions and supports integrin focal adhesion kinase (FAK) signaling, tissue stiffening, and extracellular matrix control [15].

Calcium (Ca^2+^), potassium (K^+^), and sodium (Na^+^) channels are examples of channels involved in tumor growth and metastasis [16]. Ca^2+^ is an important second messenger whose intracellular levels control several downstream signaling pathways, such as apoptosis and cell migration, functions that typically affect cancer growth [17]. In some hormone-sensitive cancers, such as breast cancer, the presence of channels in metastatic cells is regulated by positive feedback mechanisms, induced by hormone action [18]. Therefore, those ion channels represent a promising target for tumor treatment [19].

TRP channels are ionic channels permeable to monovalent and divalent cations, with a conserved structure and a higher selectivity for K^+^, Na^+^, and Ca^2+^ [20]. They have a crucial role in several pathologies, including metabolic, cardiovascular, and cancer diseases [21,22]. Recently, in a study evaluating transcriptomic and genomic alterations in TRP genes across more than 10,000 patients, it was found that 27 of 28 TRP genes are correlated with at least one hallmark of cancer in 33 different tumor types [23]. Furthermore, antagonists and agonists of TRPs have been used in association with chemotherapy in many tumor models, although side effects due to a lack of tissue specificity were observed [24,25,26,27,28].

Until now, altered levels in the function of TRP proteins in cancer have been reported, rather than mutations in the TRP genes [29]. Depending on the stage of the cancer, decreased or increased levels of the expression of the normal TRP protein can be detected [30]. Thus, these proteins could represent important markers for predicting tumor progression and, consequently, potential therapeutical targets [31,32,33,34,35]. Changes in TRP channel expression have also been associated with the staging of tumor progression [12,36,37,38]. In this review article, we summarize the latest research on the involvement of different TRP channels in cellular processes, such as proliferation, differentiation, migration, invasion, and angiogenesis, in different cancer subtypes.

## 2. TRP Channels

The TRP channel family is grouped into seven main subfamilies: ankyrin (with only one representative, TRPA1), canonical (TRPC1–7), melastatin (TRPM1–8), mucolipins (TRPML1–3), non-mechanoreceptor potential C (NOMP-like, TRPN1), polycystins (TRPP1–5), and vanilloid (TRPV1–6) [39] (Figure 1). Except for TRPN1, which has only been detected in fruit flies and zebrafish [3,6], the other TRP channels have also been detected in mammals. The mammalian TRP superfamily comprises 28 channels with a conserved primary structure that consists of six transmembrane domains (S1–S6) containing carboxy and amino terminal regions located on the intracellular side with the pore-forming loop located between S5–S6. The main differences between the seven TRP channel subfamilies are found in the N- and C-terminal cytosolic domains, which contain putative protein interaction and regulatory motifs [40]. TRP channels are activated in several ways: directly by endogenous agents, including diacylglycerols [41], phosphoinositides [42], eicosanoids [43], anandamide [44], and reactive oxygen species (ROS) and their byproducts [45,46], and by an unprecedented series of exogenous compounds, such as capsaicin [47], icilin [48], allicin [49], allyl isothiocyanate (AITC) [50,51], isopetasin [52], umbellulone [53,54], parthenolide [55], and acrolein [56], or indirectly, by intracellular mediators produced by the activation of G protein-coupled receptor (GPCR) or tyrosine kinases receptor [57].

TRP channels, almost ubiquitously expressed in cells and tissues, play important roles in health and disease [12,58], including neurological, cardiovascular, metabolic, pulmonary, psychiatric, and cancer disorders [28,59].

## 3. TRPs in Cancer

### 3.1. TRPA1 in Cancer

TRPA1, the only member of the ankyrin subfamily, is a polymodal channel that can be activated by a wide variety of noxious external stimuli, such as irritants, often associated with pain and inflammation, and intense cold [60,61,62,63]. TRPA1 can also be gated by several endogenously produced reactive chemical species, including oxidative stress by-products, such as ROS, reactive nitrogen (RNS), and carbonylic (RCS) species [45,62]. TRPA1 is predominantly expressed in the primary sensory neurons of the dorsal root (DRG), vagal (VG), and trigeminal (TG) ganglia, where it signals diverse pain stimuli [64,65,66,67,68]. TRPA1 expression has also been reported in non-neuronal cells, including the mouse inner ear and the organ of Corti [69], vascular endothelial cells [70], enterochromaffin cells of the respiratory tract [60,71], keratinocytes and melanocytes, synoviocytes, and dental pulp and gingival fibroblasts [72,73], as well as mast cells, epithelial, and pancreatic β cells [74,75,76,77,78,79,80,81]. More recently, the presence of TRPA1 in glial cells, such as astrocytes [82], oligodendrocytes [83], and Schwann cells [84,85], has been reported. In addition, the expression of TRPA1 has been observed in different cancer cells, including pancreatic adenocarcinoma and melanoma cells [86,87].

In cancer, TRPA1 activation in prostate tumor endothelial cells acts as a modulator of angiogenesis, since its activation promotes neovascularization, endothelial cell migration, and tubulogenesis in vitro in models of human prostate cancer [88]. TRPA1 activation in lung epithelial cancer cells (A549) can induce a decrease in cell invasion by inhibiting the cyclooxygenase-2 (COX-2)/prostaglandin E2 pathway in hypoxic cancer cells in vitro [89]. In breast and lung cancer spheroids, TRPA1 activates Ca^2+^-dependent antiapoptotic pathways by promoting ROS resistance [90]. Specifically, TRPA1 upregulated by nuclear factor erythroid 2-related factor 2 (NRF2), a transcription factor that, by encoding proteins with antioxidant and anti-inflammatory functions, promotes an adaptive process involving non-canonical oxidative stress defense and canonical ROS defense mechanisms [90]. Furthermore, TRPA1 inhibition attenuates xenograft tumor growth and increases chemosensitivity [90].

The TRPA1 protein has been identified in benign human skin lesions (dermal melanocytic nevi and dysplastic nevi), in cutaneous thin (pT1) and thick (pT4) melanomas, and in two different melanoma cell lines (SK-MEL-28 and WM266-4) [87]. In samples of skin lesion and melanoma, the presence of TRPA1 has been correlated with a progressive increase in oxidative stress and melanocytic transformation, as well as tumor severity. In vitro experiments on melanoma cell lines have shown that TRPA1 activation is associated with the release of H_2_O_2_, an observation in line with previous findings that have indicated the channel as an oxidative stress sensor and amplifier, a function that might affect tumor cells and proliferation [87].

### 3.2. TRPA1 in Cancer in Cancer Pain

Recent studies have underlined the role of TRPA1 in cancer pain. In a mouse model of cancer induced by the subcutaneous inoculation of melanoma B16-F10 cells, neuronal TRPA1 has been proposed to mediate mechanical and cold hypersensitivity and thigmotaxis behavior [91]. However, in mouse models of neuropathic pain induced by sciatic partial nerve ligation or ischemia and reperfusion, a prominent role of TRPA1 expressed in Schwann cells has been proposed [84,92]. Hematogenic macrophages recruited by increases in C-C Motif Chemokine Ligand 2 (CCL2) at sites of nerve injury generated a first burst of oxidative stress that, targeting the peripheral glial cell TRPA1, initiated a feed-forward mechanism that, via a Ca^2+^-dependent NADPH oxidase-1 (NOX1), amplified the oxidative stress to sustain pain signals [84,92]. Schwann cell TRPA1 has been similarly implicated in cancer-related pain by regulating, via a macrophage colony-stimulating factor (M-CSF), macrophage expansion and oxidative stress amplification, finally targeting neuronal TRPA1 to signal pain. In this mouse cancer model evoked by the inoculation of melanoma cells in the mouse paw, neuroinflammation and mechanical/cold hypersensitivity are maintained by a feed-forward mechanism, which requires continuous interaction between Schwann cell TRPA1 and expanded endoneurial macrophages throughout the entire sciatic nerve trunk [93].

Pain is also a recurrent symptom of cancer that becomes more frequent and debilitating in the presence of bone metastases, which are a common consequence of many primary tumors, including breast cancer [94]. A prominent role for the TRPA1 channel has also been reported in the development of mechanical hypersensitivity in a mouse model of metastatic bone cancer pain induced by the intramammary inoculation of breast carcinoma cells [95,96]. More recently, TRPA1 has been shown to be a regulator of metastatic bone cancer pain via insulin-like growth factor 1 receptor (IGF-1R) signaling in Schwann cells [97]. IGF-1, derived from osteoclast activation in osteolytic lesions caused by metastatic growth, targets its receptor expressed in Schwann cells, thus promoting an endothelial nitric oxide synthase-mediated TRPA1 activation and ROS release that, via M-CSF-mediated endoneurial macrophage expansion, sustains proalgesic responses.

### 3.3. TRPC

There are seven known members of the TRPC family, from TRPC1 to TRPC7. TRPC channels are widely expressed in many tissues and cells, including neurons, muscle cells, and epithelial cells. TRPC channels are involved in various physiological processes, such as sensory perception, smooth muscle contraction, hormone secretion, and cell migration [28]. TRPC channels can be activated by various signals, including G-protein-coupled receptors (GPCRs), receptor tyrosine kinases, and intracellular second messengers, such as diacylglycerol (DAG) and inositol trisphosphate (IP_3_) [98,99]. Upon activation, TRPC channels allow for the influx of Ca^2+^ ions into the cytoplasm, leading to an increase in the intracellular Ca^2+^ concentration. This Ca^2+^ influx triggers downstream signaling pathways and modulates the activity of various enzymes and transcription factors, ultimately influencing cell function. Aberrant TRPC channel activity has been associated with several diseases and pathological conditions, including cardiovascular disorders, neurodegenerative diseases, and cancer. Therefore, TRPC channels have emerged as potential therapeutic targets, and efforts are underway to develop drugs that modulate their activity for the treatment of these conditions.

#### 3.3.1. TRPC1

TRPC1 is expressed in various types of cancer, including breast, pancreatic, lung, and glioblastoma multiforme. Its dysregulation has been proposed as a prognostic marker for some types of cancer [100], including breast cancer, as its expression is modulated by tumor development and metastasis [101,102,103]. An increased expression of TRPC1 is has been positively correlated with epithelial–mesenchymal transition (EMT), a complex process that induces tumor cells to spread and fight apoptosis, thus conferring a more aggressive phenotype [104,105]. Recently, it has been reported that TRPC1 overexpression increases markers for an EMT-like phenotype, such as zinc finger proteins, SNAI1 (SNAIL) and SNAI2 (SLUG), and VIMENTIN, by increasing invasiveness in mouse breast cancer cell lines in vitro [106]. TRPC1 is also expressed in human lung carcinoma, and high protein levels have been correlated with cancer differentiation and proliferation [107].

Aberrant Ca^2+^ signaling has been implicated in glioma pathogenesis and cell biology influencing cell proliferation, migration, invasion, and angiogenesis [108]. TRPC1 channels, acting as Ca^2+^-permeable channels, have been shown to regulate Ca^2+^ influx in glioma cells [109]. Increased Ca^2+^ influx through TRPC1 channels activates downstream signaling pathways, leading to cell proliferation and survival [109]. A loss of function of TRPC1, mediated by pharmacological or genetic inhibition, was found to reduce the proliferation of multinucleated glioma cells, mainly due to the suppression of store-operated Ca^2+^ entry (SOCE) [109].

TRPC1 channels have been found to promote glioma cell migration through their association with focal adhesion proteins and cytoskeletal rearrangement. Furthermore, TRPC1-mediated Ca^2+^ signaling can activate proteases and matrix metalloproteinases, facilitating the breakdown of the extracellular matrix and promoting glioma cell invasion [110]. TRPC1 channels have been shown to promote the release of pro-angiogenic factors, such as vascular endothelial growth factor (VEGF), from glioma cells, thus inducing endothelial cell proliferation and migration and leading to the formation of new blood vessels within the tumor microenvironment [111].

#### 3.3.2. TRPC3

The TRPC3 protein might contribute to the development of tumor senescent phenotypes. Its downregulation in stromal cells promotes cellular senescence, sustaining inflammation and tumor growth in vivo [112]. TRPC3-mediated Ca^2+^ influx has been suggested as an endothelial cell attraction factor in prostate cancer, thus promoting angiogenesis [88,113]. TRPC3 overexpression has been detected in triple-negative breast cancer cells, and its activation induces an RAS P21 protein activator 4-mitogen-activated protein kinase (RASA4-MAPK) signaling cascade that plays a crucial functional role in preserving proliferation and resistance to apoptosis. A TRPC3 blocker attenuates proliferation, induces apoptosis, and sensitizes cell death to chemotherapeutic agents [114]. The TRPC3 channel is also highly expressed in gastric cancer specimens, and its expression is correlated with malignant progression by modulating the calcineurin B-like 2/glycogen synthase kinase-3 beta/nuclear factor of the activated T cells 2(CNB2/GSK3β/NFATc2) signaling pathway and controlling cell cycle, apoptosis, and intracellular ROS generation [115].

#### 3.3.3. TRPC5

An aberrant Wnt/β-catenin signaling cascade facilitates cell renewal, proliferation, and differentiation in several cancer types [116]. The activation of this intracellular pathway increases the production of the ATP-biding cassette, subfamily B, member 1 (ABCB1), a multidrug efflux transporter that attenuates the effect of cytotoxic drugs in cancer cells. TRPC5 was found to be overexpressed together with ABCB1 in colorectal cancer cells resistant to 5-fluorouracil (5-Fu). TRPC5 silencing inhibits Wnt/β-catenin signaling, thus reducing ABCB1 and consequently reverting resistance to 5-Fu [117]. In a similar manner, TRPC5 channel expression is increased in breast cancer cell lines together with P-glycoprotein (P-gp), another pump overexpressed by cancer cells to remove cytotoxic drugs. TRPC5 suppression reduces P-gp levels and causes a reversal of drug resistance in cells [118]. The mechanism by which TRPC5 regulates P-gp seems to be specifically controlled through the activation of the nuclear factor of activated T cells isoform c3 (NFATc3) [119]. In addition, in breast cancer cells, TRPC5 activation promotes autophagy and chemoresistance via the Ca^2+^/calmodulin-dependent protein kinase beta/adenosine monophosphate-activated protein kinase alpha/mechanistic target of rapamycin (CaMKKβ/AMPKα/mTOR) pathway [120]. It has also been shown that TRPC5 is highly expressed in human breast cancer after long-term chemotherapy treatment, and its presence has been correlated with an increase in the transcription of vascular endothelial growth factor, which, in turn, stimulates tumor angiogenesis [121]. The TRPC5 channel seems to promote metastasis in colon cancer. Colon cancer patients with a high expression of TRPC5 display poorer overall and metastasis-free survival [122]. TRPC5 overexpression, by increasing intracellular Ca^2+^ concentration and mesenchymal biomarker expression, promotes cell migration, invasion, and proliferation [122].

#### 3.3.4. TRPC6

Growing evidence has reported that the pattern of the expression of TRPC6 proteins is upregulated in several pathophysiological conditions, including cancer. TRPC6 has been found to be overexpressed in breast cancer biopsy tissues compared to normal breast tissues [123]. Human breast cancer cells in vitro also display significant levels of TRPC6 expression, and its silencing results in a significant reduction in cell growth [123]. In human hepatocellular carcinoma cells, transforming growth factor beta (TGFβ) is a mediator of motility, invasion, and metastases via the stimulation of Na^+^/Ca^2+^ exchanger 1 (NCX1), and TRPC6 activation regulates TGFβ, thus inducing the formation of a TRPC6/NCX1 molecular complex [124]. The expressions of both TRPC6 and NCX1 are markedly increased in human hepatocellular carcinoma tissues, and their expression levels positively correlate with migration, invasion, and intrahepatic metastasis [124]. An increased TRPC6 channel in cervical cancer cell lines induces cell proliferation, suggesting that channel inhibition might reduce the malignant behavior of the cancer. TRPC6 might be a new target for the prevention and treatment of cervical cancer [125].

### 3.4. TRPM

TRPM channels are a family of ion channels that regulate sensory perception, cellular homeostasis, and signal transduction. TRPM channels respond to a broad array of stimuli, including temperature, touch, pain, osmolarity, and chemical signals. They are expressed in various tissues and cell types throughout the body, highlighting their importance in numerous physiological functions. TRPM channels have gained significant attention in the field of cancer research due to their potential involvement in tumor progression and metastasis. Several members of the TRPM channel family have been implicated in cancer development and are investigated as potential therapeutic targets [126,127].

#### 3.4.1. TRPM1

TRPM1 gene expression has been identified in benign nevi, dysplastic nevi, and cutaneous melanomas, with a negative association between its presence and melanoma aggressiveness [128,129]. In contrast, TRPM1 protein expression is associated with tumor progression and survival in acral melanoma, supposedly because of the activation of Ca^2+^/calmodulin-dependent protein kinase II (CaMKII), which facilitates the binding of CaMKII with protein kinase B (AKT) and activates AKT, promoting melanoma cell colony formation, mobility, and an increase in tumor growth [130].

#### 3.4.2. TRPM2

TRPM2 is highly expressed in many human cancers (neuroblastoma, breast, gastric, lung, pancreatic, prostate cancer, squamous cell carcinoma, and T-cell leukemia), where its activation increases malignant cell survival [131]. The modulation of TRPM2 via oxidative stress in several pathological conditions has been reported [132]. One additional product derived from oxidative stress, ADP-ribose (ADPR), binds to the C- and N-termini of TRPM2, an action that results in channel activation [133,134,135].

Channel activation also modulates oxidative stress production. TRPM2 opening modulates the hypoxia-inducible transcription factor 1/2α (HIF-1/2α) signaling cascade, including proteins involved in oxidant stress, glycolysis, and mitochondrial function in neuroblastoma xenograft models [136]. TRPM2 inhibition or depletion reduces cell and mitochondria Ca^2+^ influx and decreases activity, autophagy, antioxidant response, and mitochondrial function, thus impairing tumor cell survival [136,137,138,139,140,141]. TRPM2 also has immunomodulatory functions and can influence the tumor microenvironment. TRPM2 activation in immune cells, such as macrophages and dendritic cells, affects their polarization and cytokine production, leading to a modulation of the anti-tumor immune response [142]. Additionally, TRPM2-mediated Ca^2+^ influx influences the release of inflammatory mediators that promote tumor growth and angiogenesis.

#### 3.4.3. TRPM3

TRPM3 plays pleiotropic roles in cellular Ca^2+^ signaling and homeostasis [143]. TRPM3 has been identified in several cancer types in mammals, including kidney cell carcinoma, glioma, melanoma, and melanoma-associated retinopathy (MAR) [144,145,146]. The TRPM3 channel supports the growth of clear cell renal cell carcinoma by promoting autophagy [144]. An increased expression of TRPM3 in renal cell carcinoma leads to Ca^2+^ influx, which elicits the activation of CaMKII, 5′-AMP-activated protein kinase (AMPK), and Unc-51-like autophagy-activating kinase 1 (ULK1), as well as the formation of phagophore [147].

MiR-204 is an intron micro-RNA (miRNA) located between exons 7 and 8 of the TRPM3 gene. A reduction in miR-204 induced by the higher methylation of host gene TRPM3 in gliomas can promote cell migration and enhance cell stemness [148]. It has been shown that TRPM3 interacts with the signal transducer and activator of transcription 3 (STAT3) via the activation of STAT3-suppressing miR-204 expression. Furthermore, the downregulation of miR-204 via the methylation of the promoter of its host gene TRPM3 leads to the activation of the Src-STAT3-NFAT pathway, promoting glioma stem cell invasion and stem cell-like phenotype [149].

#### 3.4.4. TRPM4

In physiological conditions, TRPM4 controls cell migration [150]. It regulates the activation of T lymphocyte and mast cells, together with the migration of dendritic and mast cells [151,152]. Under inflammatory conditions, TRPM4 is involved in vascular endothelial cell migration and ROS production [153]. TRPM4-mediated effects on cell migration are at least partially due to the activation of Rac family small GTPase 1 (Rac1-GTPase), a key regulator of cytoskeletal dynamics and cell polarity [154].

TRPM4 channel expression has been described in several cancers, including prostate [155,156,157], urinary bladder [158], cervical [159], colorectal [160,161], liver [162], and large B cell lymphoma [163]. In cancer cells, TRPM4 upregulation is associated with cancer cell migration, proliferation, and invasion. A recent study has shown that TRPM4 upregulation and its conductivity control the viability and cell cycle of colorectal cancer cells [164]. Another study revealed that TRPM4 gene defects mechanically engaged intestinal barrier integrity by depressing the generation of ROS and decreasing mucus production, thus promoting chronic bowel inflammation, a risk factor for colorectal cancer [160]. A more recent study indicated that an attenuated expression of TRPM4 is associated with the development of endometrial carcinoma and breast cancer through the hyperactivation of the phosphoinositide 3-kinase/protein kinase B/mammalian target of rapamycin (PI3K/AKT/mTOR) pathway, which regulates cell transcription, translation, migration, metabolism, proliferation, and survival [165]. In addition, within normal lymphoid tissues, including the tonsils, lymph nodes, and appendix, human normal B cells express low levels of TRPM4, while in diffuse large B cell lymphoma, a higher TRPM4 protein level has been detected, which confers significantly poorer patient outcomes [163]. Similarly, a higher level of TRPM4 protein correlates with a higher risk of recurrence following radical prostatectomy [157].

#### 3.4.5. TRPM6

TRPM6 is mostly expressed in the kidneys, distal small intestine, and colon [166]. The TRPM6 channel is permeable to magnesium (Mg^2+^), thus assuming a relevant role in epithelial Mg^2+^ transport and active Mg^2+^ absorption, especially in the gut and kidneys [167]. Hypomagnesemia is evidenced in cancer patients after cisplatin-based chemotherapies [168], and it has emerged as the most notable adverse effect of the anti-epidermal growth factor receptor (EGFR) monoclonal antibody, cetuximab, which is used widely for the treatment of advanced colorectal cancer cells [169]. The downregulation of the TRPM6 channel is present in 80% of primary tumors in colorectal cancer cells, whereas its high expression increases patient survival [170].

#### 3.4.6. TRPM8

TRPM8, also defined as a “cold receptor”, as it is activated by chemical cooling agents (such as menthol) [171], exhibits an increased expression in several cancer subtypes, including colon, breast, and prostate tumors, and it is considered to be a useful prognostic marker [172,173,174]. TRPM8 channels in cancer prognosis are associated with the modulation of cell viability, proliferation, migration, and apoptosis. For example, TRPM8 activation may regulate AMPK activity by modifying cellular autophagy to control the proliferation and migration of breast cancer cells. TRPM8 knockdown decreases basal autophagy, while TRPM8 overexpression increases basal autophagy in several mammalian cancer cell types. The activation of autophagy-associated signaling pathways for AMPK and unc-51-like kinase 1 (ULK1), as well as the production of phagophores, are part of the TRPM8 strategy for controlling autophagy [175].

TRPM8 involvement in the death and apoptosis of bladder cancer cells is due to the modulation of mitochondrial activity [176,177]. It has also been reported that, in glioblastoma, TRPM8 channels modulate the expression of apoptosis-related factors through the p38/MAPK pathway [178,179]. In colon, oral, esophageal, bladder, and breast cancers, TRPM8 seems directly involved in invasiveness and metastasis via the regulation of the epithelial–mesenchymal transition process (EMT) [173,180,181,182].

### 3.5. TRPML

The endolysosomal TRPML subfamily consists of the TRPML1, TRPML2, and TRPML3 proteins. TRPMLs share roughly 40% of amino acid sequence similarity and have important roles in ion homeostasis, membrane trafficking, exocytosis, and autophagy [183,184]. Two pore channels (TPCs) and TRPMLs are endolysosomal channels regulating the autophagy/lysosome system, which is intensely associated with both cancer progression and cancer escape from immunosurveillance [185]. Several recent reports have clearly proved an emerging role for the TRPML channel in cancer development and progression, and a clinical prognostic role has been suggested [25,185,186].

#### 3.5.1. TRPML1

TRPML1, mainly located in lysosomes, promotes cation efflux into the cytosol [187], thus controlling lysosomal storage, transportation, and pH homeostasis. TRPML1 mutations alter lysosomal storage, and lysosomal impairment is responsible for autophagy distortions. TRPML1 can also be negatively regulated by the target of rapamycin (TOR) with a consequent autophagy reduction, thus supporting a central role of TRPML1 in this process [188]. TRPML1 also regulates the exocytosis of intracellular contents via the endosomal lysosomal pathway [189,190]. Recent studies have revealed that TRPML1 is significantly increased in HRAS-positive tumors and oppositely correlated with patient prognosis. The knockdown, or selective inhibition, of TRPML1 abolishes the proliferation of cancer cells that express oncogenic HRAS [191]. An increased expression of the TRPML1 gene is also observed in melanoma cells compared to normal melanocytes, and TRPML1-deficient melanoma cells exhibit decreased survival, proliferation, and tumor growth [192].

Recently, an increased TRPML1 expression level has been correlated with poor clinical outcomes of pancreatic ductal adenocarcinoma patients [193], significantly lowering overall survival. A role of TRPML1 in pancreatic ductal adenocarcinoma progression has been further investigated by using an in vitro cell model, and the results show that the proliferation of pancreatic ductal adenocarcinoma cells is blocked by TRPML1 depletion. Parallel to this result, in a pancreatic ductal adenocarcinoma mouse model, TRPML1 was crucial for the formation and growth of tumors [193]. Altogether, these studies suggest that TRPML1 is upregulated in cancer cells to promote tumorigenesis. In line with this conclusion, another study suggests that TRPML1-mediated lysosomal exocytosis, which releases high levels of ATP to the extracellular space, promotes triple-negative breast cancer cell invasion and metastasis [194]. Conversely, another recent study proposes that TRPML1 activation in glioblastoma cell lines reduces cell viability by inducing caspase-3-dependent apoptosis [195]. Thus, the loss of TRPML1 expression is strongly correlated with a short survival in glioblastoma patients, suggesting that a reduction in TRPML1 expression represents a negative prognostic factor in glioblastoma patients [195].

#### 3.5.2. TRPML2

While TRPML1 is mainly localized in late endosomes and lysosomes in all tissues, TRPML2 is primarily expressed in myeloid and lymphoid cell lineages as recycling endosomes [196]. Due to its higher expression in immune cells compared to other endolysosomal ion channels, a role for the channel in innate immune responses has been postulated [197,198]. TRPML2-knockout mice display an impaired recruitment of peripheral macrophages in response to inflammation, suggesting a potential defect in the immune response.

A link between TRPML2 expression and cancer has been investigated in different tumor types. Overall, a pro-tumorigenic role of TRPML2 has recently been proposed [199]. TRPML2 silencing decreases proliferation and cell viability by abolishing AKT/ERK1/2 phosphorylation and provokes apoptosis in glioma cell lines. Moreover, a role for TRPML2 in prostate cancer has recently been reported [200]. Via the regulation of the interleukin-1 beta/factor nuclear kappa B (IL-1β/NF-κB) pathway, TRPML2 activation promotes cancer cell proliferation, migration, and invasion.

#### 3.5.3. TRPML3

Differing from TRPML1, which is ubiquitously expressed in all tissues, TRPML3 is mainly expressed in specific organs, such as the kidneys, lungs, back skin, and thymus [201,202]. TRPML3 was discovered in plasma membrane and multiple intracellular compartments, including autophagosomes, early and late endosomes, and lysosomes, where channel activation is involved in autophagy regulation [203]. Although its role in cancer has been poorly explored, a detailed analysis in ‘The Cancer Genome Atlas’ (TCGA) revealed that the downregulation of TRPML3/MCOLN3 is associated with a relatively better survival in several types of cancers, including adrenocortical, breast invasive, uterine corpus endometrial, kidney renal clear cell, and kidney papillary cell carcinomas, colon, lung, lung squamous cell, rectal, and stomach adenocarcinomas, pheochromocytoma, and paraganglioma, thymoma, and uterine carcinosarcoma [204].

### 3.6. TRPP

The TRPP family consists of three members: TRPP2, TRPP3, and TRPP5. TRPP2 has been demonstrated to be an active coordinator in TRP channel heteromerization. The physical interaction of TRPP2, also known as polycystin-2 (PKD2), with polycystin-1 (PKD1) was identified 26 years ago [205]. A study reported that PKD1 is present in stem cells of variable origins. The overexpression of PKD1 enhances the cell mobility and differentiation in umbilical-cord-blood-derived stem cells [206]. Functionally, TRPP2 is involved in the regulation of smooth muscle contraction, cell proliferation, and mechanical sensation [207]. It has been reported that TRPP2 activation enhances the metastasis of laryngeal squamous cell carcinoma, regulating the EMT [208]. The silencing of TRPP2 significantly suppresses ATP-induced Ca^2+^ release, wound healing, and cell invasion, which collaboratively diminishes the SMAD family member 4 (Smad4), STAT3, SNAIL, SLUG, and TWIST expression.

### 3.7. TRPV

TRPV channels are a group of ion channels that play crucial roles in sensory signaling and are involved in a wide range of physiological processes. These channels are named after their founding member, vanilloid 1 (TRPV1), which deorphanized the so-called receptor for capsaicin, the spicy ingredient of chili peppers [209,210]. TRPV channels are widespread in various tissues throughout the body, including neurons, epithelial cells, and immune cells. They are non-selective cation channels, allowing for the influx of Ca^2+^, Na^+^, and K^+^ across the cell membrane. TRPV channels are characterized by their sensitivity to multiple physical and chemical stimuli, including temperature, pH, mechanical stress, and various endogenous and exogenous compounds.

TRPV channels have been increasingly involved in cancer development and progression. Emerging evidence suggests that these channels play significant roles in various aspects of cancer biology, including tumor cell proliferation, migration, invasion, angiogenesis, and resistance to therapy. The dysregulation of TRPV channels in cancer suggests their potential as therapeutic targets. Modulating the activity of TRPV channels could help to inhibit tumor growth and metastasis and enhance the efficacy of existing therapies.

#### 3.7.1. TRPV1

The TRPV1 channel is the selective molecular target of the vanilloid capsaicin, but it can also be gated by acid (pH < 6.5), ethanol, and heat [209,211,212]. Although TRPV1 is mainly localized to the plasma membrane, it can also be detected in the endoplasmic and sarcoplasmic reticulum [213,214], providing a pathway for the release of Ca^2+^ from these intracellular stores. The capsaicin-induced stimulation of TRPV1 triggers the apoptosis of human urothelial cancer cells via the activation of the ataxia telangiectasia mutated/CHK2/p53DNA damage response and Fas/CD95-mediated apoptotic pathways [215]. In tumor cells, proliferation, invasion, and metastasis are controlled by Ca^2+^ signaling. TRPV1 is functionally expressed in human esophageal squamous cells, and thermo-TRPVs might play an important role in the development of esophageal squamous cells [216]. It has been reported that TRPV1 overactivation supports the proliferation and/or migration of esophageal squamous cells.

An elevated TRPV1 expression has been proven in squamous cell carcinoma of the human tongue, lung cancer, and breast cancer [217,218,219]. TRPV1 exceptionally suppresses the development of gastric cancer through a novel Ca^2+^/CaMKKβ/AMPK pathway, and its downregulation has been associated with poor survival in human gastric cancer patients. Thus, TRPV1 upregulation and its downstream signaling may represent promising targets for gastric cancer prevention and therapy [220]. Little is known about TRPV1 and cancer-induced chemosensitivity.

#### 3.7.2. TRPV2

While TRPV1 expression is primarily localized to the plasma membrane [221], TRPV2 is found in intracellular membranes [222]. TRPV2 is implicated in the signaling pathways that mediate cell survival, proliferation, and metastasis. In leukemia and bladder cancer, the oncogenic activity of TRPV2 relates to a different expression profile of the receptor. It can be overexpressed in cancerous cells, increasing tumor aggressivity, and its silencing or blocking can trigger apoptosis and cell cycle arrest [223]. Oxidative stress generated by TRPV2 in human hepatoma cell lines induces cell death, involving the inhibition of pro-survival signaling proteins and the activation of pro-death signaling proteins [224]. The higher expression of TRPV2 in gastric cancer patients has been proposed as a prognostic biomarker and potential therapeutic target [225]. TRPV2 expression has been evaluated in epidermal melanocytes, two human malignant melanoma, and two metastatic melanoma cell lines [226]. TRPV2-mediated melanoma cell death via channel activation favors the antitumor process.

In multiple myeloma patients, TRPV2 overexpression is associated with bone tissue damage and a poor prognosis. A loss or inactivation of TRPV2 also increases glioblastoma cell proliferation and provokes resistance to CD95-induced apoptotic cell death [145]. TRPV2 mediates cell adhesion, migration, and invasion by stimulating adrenomedullin in prostate and urothelial cancer [227]. Adrenomedullin induces prostate and urothelial cancer cell migration and invasion through TRPV2 translocation to the plasma membrane and an ensuing increase in the resting Ca^2+^ level. In prostate cancer, TRPV2 overexpression is also correlated with castration-resistant phenotype and metastasis [228]. Recent analyses have demonstrated that high expressions of TRPV2 and TRPM4 are negatively correlated with the prognosis of uveal melanoma patients [165]. TRPV2 overexpression is also linked to high relapse-free survival in triple-negative breast cancer, where the reverse is found in patients with esophageal squamous cell carcinoma or gastric cancer. Overall, these findings validate TRPV2 as a potential candidate for a cancer biomarker and future therapeutic target [223].

#### 3.7.3. TRPV3

TRPV3 is a Ca^2+^-permeable nonselective cation channel broadly expressed in skin keratinocytes, along with oral and nasal epithelia [229]. Although the role of TRPV3 in cancer has not been extensively studied, emerging evidence suggests its potential implications in cancer development and progression. Several studies have reported altered expression patterns of TRPV3 in various cancer types, indicating its possible involvement in oncogenic processes. In some cancers, such as non-small lung cancer, melanoma, squamous cell carcinoma, and breast cancer, TRPV3 expression has been found to be upregulated. An increased TRPV3 expression has been associated with cancer cell proliferation, survival, and invasion [230,231]. The activation of TRPV3 has been shown to promote cancer cell growth by triggering the intracellular signaling pathways involved in cell proliferation, such as the ERK1/2 pathway. Additionally, TRPV3 activation can induce the expressions of genes associated with cancer progression and metastasis, including matrix metalloproteinases (MMPs) and VEGF.

#### 3.7.4. TRPV4

The TRPV4 channel is a non-selective cation channel that is extensively expressed in several tissues, where it acts as a molecular sensor and a transducer that regulates a variety of functional activities [232,233,234,235,236]. Like other channels, the modulation of TRPV4 channel expression is observed to be closely related to tumor formation progression and metastasis. TRPV4 is overexpressed in colorectal, lung, and gastric cancer cells, but in other tumors, including prostate, skin, and esophageal cancer cells, TRPV4 channel expression appears to be normal [237,238]. The latest findings indicate that TRPV4 induces apoptosis via p38 MAPK in human lung cancer cells. TRPV4 overexpression in human lung cancer cell lines induces cell death and inhibits cell proliferation and migration. The inhibition of p38 MAPK reduces TRPV4’s effects on the cell proliferation, apoptosis, and migration of those cells. Collectively, this can imply that TRPV4 is a candidate target for human lung cancer therapy [239].

An abnormal TRPV4 expression is linked to gastric, liver, pancreatic, colorectal, lung, and breast cancers [237,240]. The upregulation of TRPV4 has been identified in breast cancer cell lines with the potential to metastasize, and its expression appears to increase with tumor grade and size, and to correlate with poor survival [241,242]. Furthermore, TRPV4 has been linked to cell proliferation through the CaMKII pathway and the regulation of apoptosis in distinct cancer models [243,244]. According to recent studies, tumor growth and metastasis are significantly increased in a syngeneic Lewis lung carcinoma tumor model of endothelial-specific TRPV4 knockout (TRPV4-ECKO) mice compared to wild-type mice [245]. This tumor growth is accompanied by increased tumor angiogenesis and metastasis compatible with the abnormal leaky vessels observed. Mechanistically, in TRPV4-ECKO mouse tumors, proteins that are related to endothelial structure and cell proliferation, such as vascular endothelial growth factor receptor 2 (VEGFR2), p-ERK, and MMP-9 expression, are increased. Thus, endothelial TRPV4 emerges as a subtle modulator of vascular integrity and an inhibitor of tumor angiogenesis, given that the deletion of TRPV4 promotes tumor angiogenesis, growth, and metastasis [245]. TRPV4 expression is higher in breast metastatic lesions compared to normal breast tissue and invasive ductal carcinomas, and its expression increases with tumor grade and size [246]. Conversely, TRPV4 is markedly downregulated in keratinocytes in the premalignant lesions of non-melanoma skin cancer, such as solar keratosis and Bowen’s disease, and in basal and squamous cell carcinoma [247]. The suppression of TRPV4 expression in keratinocytes has been correlated with the increase in cytokines and prostaglandins within the tumor milieu [247].

#### 3.7.5. TRPV5

TRPV5 is a Ca^2+^-selective ion channel widely expressed in many tissues, including urinary bladder and kidney, where it acts as a gatekeeper of active Ca^2+^ reabsorption [248,249]. Altered TRPV5 expression has been identified among the different renal cell carcinoma histopathological subtypes. An altered vitamin D receptor expression may be associated with renal cell carcinoma carcinogenesis via TRPV5/6 [250].

Ca^2+^ deficiency triggers abnormal colonic growth and increases colon cancer risk with well-known mechanisms [251]. The parathyroid glands play an overall regulatory role in systemic Ca^2+^ homeostasis. There was a presence of TRPV5 and TRPV6 in sporadic parathyroid adenomas and normal parathyroid glands co-localized with Ca^2+^-sensing receptors on the membrane surface, although immunoreactivity was present in the cytosol and around the nuclei. An increased expression of both channels in adenoma compared to normal glands implies a relationship between cell Ca^2+^ signaling and pathological processes [252]. A recent study showed that low Ca^2+^-induced IGF signaling is mediated by TRPV5-associated membrane depolarization. These results disclose a novel signaling mechanism that results in abnormal epithelial proliferation associated with Ca^2+^ deficiency [253]. A reduced TRPV5 expression in tumor tissues is detected in non-small-cell lung cancer patients and associated with a shorter median survival time after surgical resection. The combined expression of TRPV5 and TRPV6 in tumor tissues exhibited promising prognostic value in non-small-cell lung cancer patients [254].

#### 3.7.6. TRPV6

TRPV6 is a membrane Ca^2+^ channel widely expressed by the epithelial tissues of many tissues, such as the intestines, kidneys, placenta, epididymis, and exocrine glands [255]. The expression of the TRPV6 gene is remarkably upregulated in several human malignancies, including the most common cancers: prostate and breast cancer [34,256]. TRPV6 appears to be expressed in various cancer cell lines, but a direct identification of the TRPV6 protein using mass spectrometry has only been shown in the human breast cancer cell line, T47D [257], and in the human lymph node prostate cancer cell line, LNCaP [258].

Mechanistically, TRPV6-mediated Ca^2+^ influx maintains the inactive state by targeting IGF-mediated AKT-TOR and ERK signaling. A recent study showed that, in zebrafish epithelia and human colon carcinoma cells, TRPV6 diminishes intracellular Ca^2+^ levels and activates protein phosphatase 2A (PP2A), which downregulates IGF signaling and endorses the inactive state. This suggests that TRPV6 mediates a constitutive Ca^2+^ influx into epithelial cells to continuously suppress growth factor signaling and maintain the quiescent state [259]. In addition, TRPV6 activates NFATc2 by increasing the nuclear factor of activated T cells 2 interacting protein (NFATc2IP) phosphorylation, and CDK5 may be the candidate for performing this phosphorylation. Consequently, activated NFATc2 escalates breast cancer metastasis by upregulating ADAM metallopeptidase with thrombospondin type 1 motif 6 (ADAMTS6) expression. These findings indicate that TRPV6 increases NFATc2 transcriptional activity by targeting NFATc2IP phosphorylation, which finally upregulates ADAMTS6 expression to stimulate breast cancer metastasis [260]. One study has revealed a co-expression pattern between TRPV5 and TRPV6 channels in the human myeloid leukemia cell line K562, which proposes that the channel interaction contributes to intracellular Ca^2+^ signaling [261] (Table 1).

## 4. Translational Approaches

First, promising clinical studies were directed to test novel TRP channel modulators for treating inflammation and pain. However, the improved understanding of the role of these channels in different pathophysiological functions has indicated their major involvement in specific phases of cancer progression, thus suggesting that they are promising as potential targets.

To date, there are only two clinical trials for TRPA1 related to cancer. An ongoing interventional study (NCT05024383) aims to investigate the role of the TRPA1 channel in oral cancer pain. In particular, the study evaluates the mechanical (von Frey testing) and chemical (capsaicin and AITC) response in oral cancer patients and compares their sensitivities to healthy subjects. Another interventional study (NCT04923412) aims to investigate TRPA1 and TRPV1 expression in non-small cell lung cancer (NSCLC) patients before and after surgery to quantitatively measure injuries of the vagus nerve during mediastinal lymph node dissection. A lobectomy with lymph node dissection represents the standard surgical method. However, about 60% of patients experience postoperative chronic cough during the first year after surgery. In this trial, the investigators provide a new basis for safe and effective new surgical techniques, as well as possible new biomarkers (TRPA1 and TRPV1) related to postoperative cough and pulmonary complications.

Regarding TRPV1, two cancer clinical trials have been reported so far. A phase III study (NCT04572776) is currently investigating the role of TRPV1 in intractable cancer pain through a single epidural administration of resiniferatoxin vs. the standard of care. A small prospective study (NCT02666976) evaluates the modulation of inflammatory biomarkers, including TRPV1, in patients affected by gastrointestinal subepithelial tumors. One observational study (NCT05507879) evaluates the role of TRPC6 as a predictive biomarker of chemotherapy-related cardiac toxicity in patients with breast cancer. A small interventional phase I clinical trial (NCT01578564) tests the safety and tolerability of a TRPV6 inhibitor in subjects with advanced ovarian cancer or other cancers known to overexpress the TRPV6 channel.

## 5. Conclusions

Over the years, researchers have revealed that TRP channels play crucial roles in cellular homeostasis and represent an efficient and timely interface between the environment and the body. The dysregulation of TRP channels has been associated with numerous cancer types, highlighting their potential as therapeutic targets [22,23].. The role of TRPA1 in sustaining chronic cancer pain in rodent models is supported by robust data [92,94]. Emerging evidence suggests that TRP channels are implicated in cancer progression and metastatic processes. TRP channels have been also proposed as biomarkers for cancer diagnosis and prognosis, providing valuable insights into patient stratification and personalized medicine approaches [23,24,25,26,27,28]. However, due to the broad tissue distribution and multiple functions of TRPs in the same types of cancer and healthy tissues, their inhibition may result in systemic toxicity. Thus, the identification of the specific TRP channel and its selective cellular localization implicated in a given pro-oncogenic function is of paramount importance for the development of safer and better anticancer drugs. New insights into tumor biology driven by TRP channels clearly require further effort and dedicated study.

## Figures and Tables

**Figure 1 biomolecules-13-01557-f001:**
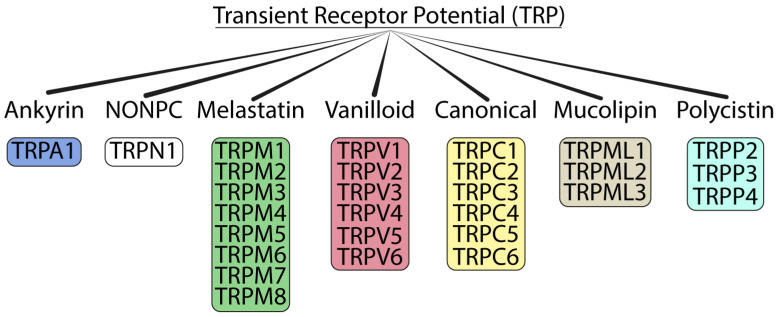
Tree of the TRP ion channel superfamily.

**Table 1 biomolecules-13-01557-t001:** TRP channels and cancer types.

TRP Channel		Cancer Type	Cell Type	Effect	Relative Expression	Reference
TRPA1		Pancreatic adenocarcinoma	Panc-1 cells	Not exact outcome	Upregulated	[86]
		Melanoma	SK-MEL-28 and WM266-4	Amplifies the oxidative stress signal that affects tumor cell survival and proliferation	-	[87]
		Prostate cancer	Prostate tumor endothelial cells (PTEC)	Angiogenic factor	Upregulated	[88]
		Breast cancer	Invasive ductal breast carcinoma	Upregulates Ca^2+^ dependent anti-apoptotic pathway	Upregulated	[90,96]
		Lung cancer	Human pulmonary epithelial A549 cells	Suppresses hypoxia-induced COX-2	Activated	[89]
		Bone cancer	4T1 breast cancer cell line	Induces production of inflammatory substances	Activated	[95]
TRPC		Breast cancer	Human breast ductal adenocarcinoma (hBDA), Small breast epithelial mucin (SBEM)	-	Upregulated	[101,102,103,106]
	TRPC1	Lung carcinoma	Non-small cell lung cancer (NSCLC), A549 cell line	Enhances Ca^2+^ influx and proliferation	Upregulated	[107]
		Glioblastoma Multiforme	D54MG (GMB cell line)	Reduces Ca^2+^ influx and proliferation	-	[100]
		Glioma	Malignant glioma cell line, U-87 MG cells	Hypoxia-induced VEGF expression	Unchanged	[108,110,111]
	TRPC3	Prostate cancer	Human prostate tumor-derived ECs (TECs)	Endothelial prostate cancer attraction factor	Upregulated	[88,113]
		Breast cancer	Triple-negative breast cancer (TNBC) cell line MDA-MB-231	Ca^2+^-promoted Ras-MAPK pathway suppressor	Upregulated	[114]
		Gastric cancer	Clinical GC specimens	Regulates ROCE-AKT/GSK3β-CNB2/NFATc2 cascade for cell cycle checkpoint, apoptosis, and intracellular ROS production	Upregulated	[115]
	TRPC5	Colorectal cancer	CRC cell line HCT-8 and Lo Vo	Increases nuclear β-catenin accumulation for chemotherapy resistance	Upregulated	[117]
		Breast cancer	MCF-7, T47D, and MDA-MB 231 cells	Promotes drug resistance via CaMKKβ/AMPKα/mTOR	Upregulated	[118,119,120,121]
		Colon cancer	SW620, RKO, SW1116, HT29, and HCT116 cell lines	Promotes tumor metastasis via the HIF-1α-Twist	Upregulated	[122]
		Breast cancer	Breast cancer biopsy tissue, MCF-7 and MDA-MB-231 cell line	-	Upregulated	[123]
	TRPC6	Hepatocellular carcinoma	HepG2 and Huh7 cell lines	Mediates TGFβ-driven cell migration and invasion	Upregulated	[124]
		Cervical cancer	HeLa and SiHa cell lines	-	Up-regulated	[125]
TRPM	TRPM1	Acral melanoma	Tumor specimens	Promotes tumor progression and malignancy by activating Ca^2+^CaMKIIδ/AKT pathway	Upregulated	[130]
	TRPM2	Neuroblastoma	TRPM2-depleted SH-SY5Y neuroblastoma cells	Modulates cell survival through mitochondrial ROS	Downregulated	[136]
		T-cell leukemia	4T1, LLC, 4T07, and 168FARN cancer cell line	-	Upregulated	[142]
	TRPM3	Renal cell carcinoma	786-O and A498 cell lines	The von Hippel–Lindau tumor suppressor (VHL) represses TRPM3 through miR-204	-	[144,147]
		Glioma	LN382T and SNB19 cell line	Loss in glioma	Downregulated	[145]
		Melanoma	Retinal	-	Upregulated	[146]
		Melanoma-associated retinopathy (MAR)	Retinal Pigment Epithelium (RPE)	-	Upregulated	[146]
	TRPM4	Prostate cancer	PC3 and LnCaP cell lines	Induces the expression of Snail1 gene and EMT	Upregulated	[155,156,157]
		Urinary bladder cancer	Samples from patients diagnosed with bladder cancer	-	No difference	[158]
		Cervical cancer	HT-3, ME-180, CaSki, MS751, C-4I, C-33A, SW756, HeLa, and SiHa cell lines and primary tumor specimens	Enhances tumorigenesis	Upregulated	[159]
		Colorectal cancer	Tumor specimen from CRC patients	Contributes to proliferation and invasion of tumor cells	Upregulated	[160,161,164]
		Hepatocellular carcinoma	-	Contributes to proliferation, adhesion, and migration of tumor cells	-	[162]
		Large B cell lymphoma	Diffuse large B cell lymphoma (DLBCL)	-	Upregulated	[163]
		Endometrial carcinoma	Specimens from patients with EC	Strongly associated with pro-cancer signaling pathways such as P13K-AKT-Mtor	Upregulated	[165]
		Breast cancer	MDA-MB-231, MCF7, MCF10A, and T47D	-	Upregulated	[165]
	TRPM6	Colorectal cancer	Human colon cancer samples	-	Downregulated	[169,170]
	TRPM8	Colon cancer	Tumor samples from colon cancer patients and CT26 cell line	Promotes colon cancer liver metastasis via Akt/GSK3	Upregulated	[180]
		Breast cancer	Breast cancer cell line MCF-7	Channel expression is regulated by estrogen receptor-α	Upregulated	[172,173]
		Prostate tumor	Human neoplastic prostatic tissue, BPH-1, LNCaP, C4-2B, VCaP, and NCI-H660 cell lines	Promotes hypoxic growth adaptation of cancer cells via RACK1-mediated stabilization of HIF-1α	Upregulated	[174,262]
		Bladder cancer	Human bladder cancer cell line T24	Induces mitochondrial membrane depolarization and cell death	Upregulated	[176,177]
		Glioblastoma	Human glioblastoma cell line U251	Contributes to survival, proliferation, apoptosis, and local tumor invasion	Upregulated	[178,179]
		Esophageal cancer	Human esophageal cancer cell line: EC109, KYSE-150, TE1, and TE10. Also, tissues from patients diagnosed with esophageal cancer	Mediates activation of the calcineurin-NFATc3 signaling pathway and PD-L1 expression	Upregulated	[182,262]
TRPML	TRPML1	Melanoma	Melanoma specimen from patients	Promotes protein homeostasis in melanoma cells by negatively regulating MAPK and mTORC1	Upregulated	[192]
		Pancreatic ductal adenocarcinoma	Fresh samples from patients with PDAC and human pancreatic cancer lines PANC-1 and BxPC-3	-	Upregulated	[193]
		Breast cancer	MCF10, MDA-MB-231, MCF7, Hs 578 T, and SUM159PT breast cancer cell line	Regulates cancer development by promoting mTORC1 and purinergic signaling	Upregulated	[194]
		Glioblastoma	Biopsies from patients and T98 and U251 cell lines	Releases intracellular Ca^2+^ autophagy and apoptotic cell death	Downregulated	[195]
	TRPML2	Glioma	T98 and U251 cell lines	Promotes cancer progression	Upregulated	[199]
		Prostate cancer	Human PCa cell lines PC-3, DU145, and LNCaP	Promotes cancer progression via IL-1β/NF-κB pathway	Upregulated	[200]
	TRPML3	Adrenocortical, Breast invasive, Uterine corpus endometrial, Kidney renal clear cell, and Kidney papillary cell carcinomas, Colon, Lung, Lung squamous cell, rectal, and Stomach adenocarcinomas, Pheochromocytoma, and Paraganglioma, Thymoma, and Uterine carcinosarcoma	-	Regulates autophagy and autophagosome formation	Downregulated	[204]
TRPP		Laryngeal squamous cell carcinoma	Human LSCC tissue and Hep 2 cell line	TRPP2 siRNA significantly decreased Smad4, STAT3, SNAIL, SLUG, and TWIST expression	Upregulated	
TRPV	TRPV1	Bladder cancer	Human urothelial cell lines J82, EJ, and TCCSUP	Induces Fas/CD95-mediated intrinsic ad extrinsic pathways	Downregulated	
		Esophageal squamous cell carcinoma	Human ESCC cell lines Eca109 and TE-1	Promotes cellular proliferation and migration	Upregulated	[216]
		Lung cancer	A549 cell line	Induces chemoresistance by upregulation of ABCA5 drug transporter gene and increases IL-8 signaling and cell survival	Upregulated	[218]
		Breast cancer	Breast cancer tissues from patients and SUM149PY cell line	Enhances apoptosis	Upregulated	[219]
		Gastric cancer	Human primary GC tissues	Suppresses GC development through Ca^2+^/CaMKKβ/AMPK	Downregulated	[220]
		Oral cancer	Fresh tissues from patients with tongue SCC or epithelial leukoplakia	-	Upregulated	[217]
	TRPV2	Leukemia	K562, U937, and THP-1 cell lines	Promotes cell survival and growth	Upregulated	[223]
		Hepatocellular carcinoma	HepG2 and Huh7 cell lines	Promotes cytotoxicity of H_2_0_2_- via activation of p38 and JNK1	Upregulated	[224]
		Gastric cancer	Gastric Cancer patients	Maintains intracellular Ca^2+^ at low concentration and cell resistance to apoptosis	Upregulated	[225]
		Malignant Melanoma	A2058 and A375 cell lines	Regulates AKT pathway and antitumor process	Upregulated	[226]
		Glioblastoma	GBM patient tumor	Reduces cell proliferation and enhances Fas-induced apoptosis via ERK	Upregulated	[145]
		Prostate and urothelial cancer	Human PCa cell line PC-3 and urothelial cancer cell line T24/83	Mediates adrenomedullin stimulation, cell adhesion, migration, and invasion	Upregulated	[227,228]
		Uveal Melanoma	Patient samples and A2058 cell line	Mediates cell invasion via Ca^2+^-sensitive protease calpain	Upregulated	[165]
		Esophageal squamous cell carcinoma	TE5, TE8, TE19m, and TE15 cell line	Regulates cancer progression via WNT/β-catenin or basal cell carcinoma signaling	Upregulated	[223]
		Breast cancer	MCF-7, BT-474, and MDA-MB-231 cell line	-	Upregulated	[223]
	TRPV3	Non-small lung cancer	Specimens from patients with NSCLC and A549 and H1299 cell lines	Modulates cell cycle arrest via reduction of Cyclin A, D1, and E	Upregulated	[230,231]
		Squamous cell carcinoma	A549 and H1299 lung cancer cell lines	Modulates intracellular Ca^2+^ influx and facilitates phosphorylation of CaMKII	Upregulated	
	TRPV4	Colorectal cancer	Specimens from patients with colon adenocarcinoma	Regulates Ca^2+^ homeostasis for cell proliferation, differentiation, apoptosis, and migration	Upregulated	[237]
		Lung cancer	A549 and H460 human lung cancer cell line	Induces apoptosis via p38 MAPK	Upregulated	[239]
		Esophageal cancer	Primary epithelial cells	Reduces ATP, cell proliferation, and migration	Downregulated	[238]
		Breast cancer	Patient samples and 4T07 cell line	Activates AKT and downregulates E-cadherin	Upregulated	[241,242,246]
		Pancreatic cancer	Fresh tissues from patients with PDAC	Modulates Ca^2+^ mobilization and mitochondrial depolarization	Upregulated	[240]
		Lewis lung carcinoma	Primary mouse endothelial cells and endothelial-specific TRPV4 knockout mice line	Prevents tumor growth and metastasis via modulation of tumor angiogenesis	Downregulated	[245]
	TRPV5	Renal cell carcinoma	Fresh-frozen primary tumor from renal cell carcinoma patients	Inhibits tumor growth, angiogenesis, and metastasis via Vitamin D receptor	Downregulated	[250]
		Non-small lung cancer	Fresh tumor samples collected from Non-small lung cancer patients	-	Downregulated	[254]
		Colon cancer	Colon Adenoma, adenocarcinoma	Facilitates the onset and progression of cancer	Upregulated	[251,252]
	TRPV6	Breast cancer	Breast cancer cell line T47D	Promotes cancer cell growth or survival	Upregulated	[256,257,260]
		Prostate cancer	Prostate tissue from patients with prostate cancer	Increases apoptosis resistance and proliferation of cancer cells	Upregulated	[34]
		Colon carcinoma	Human LoVo colon carcinoma cell	Promotes proliferation by decreasing intracellular Ca^2+^ and activation of IGF	Downregulated	[259]
		Acute myeloid leukemia	Human myeloid leukemia K562 cells	Promotes Ca^2+^ homeostasis and proliferation of cancer cells	Upregulated	[261]

## Data Availability

Not applicable.

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
