# Peer review of "TRP Channels in Cancer: Signaling Mechanisms and Translational Approaches"

_biomolecules, 2023, doi:10.3390/biom13101557_

Round 1

Reviewer 1 Report

This is a useful aggregation of reports detailing the involvement of various TRP channels in various aspects of cancer development and progression. There are two major points to be addressed, and several minor comments.

Major.

#1. One clear deficiency of the review is the lack of discussion of TRP ligands, and their role in either clinical trials or experimental assays as to their anticancer effects. While the review succeeds in addressing the first part of the title (‘signaling mechanisms’), there is a complete absence of discussion of the second part (‘translational approaches’). There needs to be a section discussing therapy with TRP ligands.

#2. The Table is not very helpful. It simply lists TRP channels and different types of cancer. The table needs qualification as to how these channels are involved in the different cancers. Do the studies show positive or negative linkage to cancer outcomes? Are the channels up or down regulated in these associations? Currently the table provides rather limited information, and arguably little new information compared with previous reviews on this topic.

Minor

Line 19. Change ‘have shown their implication’ to ‘are involved’.

Lines 54-66.  This is controversial as to whether TRPs are true ‘primary’ mechanosensors, like Piezo. I think this qualification should be introduced that perhaps TRP as indirect mechanosensors.

Line 75. Not true that all TRPs are Ca2+ permeable.

Line 93. More than 50 subtypes? What are the authors defining as a subtype?

Line 300-1. There are ADPR binding sites in the NH2 terminus of TRPM2 also. Cite appropriate refs.

OK.

Author Response

Dear Editor,

We thank you and the reviewers for the helpful comments and critiques. We have revised the manuscript in response to the comments. The changes are highlighted in blue in the revised manuscript. We have also prepared a point-by-point reply to all the issues that were raised by the reviewers.

Please do not hesitate to contact us should you require further information.

Sincerely,

Romina Nassini

Reviewer 1:

This is a useful aggregation of reports detailing the involvement of various TRP channels in various aspects of cancer development and progression. There are two major points to be addressed, and several minor comments.

Major.

#1. One clear deficiency of the review is the lack of discussion of TRP ligands, and their role in either clinical trials or experimental assays as to their anticancer effects. While the review succeeds in addressing the first part of the title (‘signaling mechanisms’), there is a complete absence of discussion of the second part (‘translational approaches’). There needs to be a section discussing therapy with TRP ligands.

We thank the reviewer for the suggestion. An updated topic regarding clinical trials has been added in a new paragraph (Page 18 Line 624).

#2. The Table is not very helpful. It simply lists TRP channels and different types of cancer. The table needs qualification as to how these channels are involved in the different cancers. Do the studies show positive or negative linkage to cancer outcomes? Are the channels up or down regulated in these associations? Currently the table provides rather limited information, and arguably little new information compared with previous reviews on this topic.

Thank you for the suggestion. The table has been now improved by adding more information regarding TRP actions on different types of cancer (Pages 13).

Minor

Line 19. Change ‘have shown their implication’ to ‘are involved’.

The sentence has been modified (Page 1 Line 19)

Lines 54-66.  This is controversial as to whether TRPs are true ‘primary’ mechanosensors, like Piezo. I think this qualification should be introduced that perhaps TRP as indirect mechanosensors.

The sentence has been rephrased (Page 2 Lines 56)

Line 75. Not true that all TRPs are Ca2+ permeable.

The sentence has been modified (Page 2 Lines 75)

Line 93. More than 50 subtypes? What are the authors defining as a subtype?

The sentence has been corrected (Page 2 Line 94).

Line 300-1. There are ADPR binding sites in the NH2 terminus of TRPM2 also. Cite appropriate refs.

The sentence has been corrected and the refs (Refs number 133-135) about the ADPR in the N-terminus have been added (Page 7 Line 304).

Reviewer 2 Report

Marini et al submitted a review entitled « TRP channels in cancer : signalling mechanisms and translational approaches ».

Major points :

Although this review is well documented, it is presented as a catalog of the role of TRP channels in cancer. This review fails to deliver on the state of research into the use of TRP channel modulators in oncology as suggested in the title and abstract.

To improve the interest of the review, the authors have to :

-          add summary diagram(s) of TRP-activated transduction mechanisms involved in cancer cell proliferation, migration or drug response.

-          Describe and name TRP modulators and report clinical trials in progress and those with known results.

A discussion concerning  the involvement of multiple TRP channels in the same type of cancer (breast, prostate, lung), would be interesting to bring to the reader.

Minor points

A schematic picture of the TRP channel subfamiy structures would be appreciated.

Page 4 line 184 : TRPC cells instead of TRPC channels

Author Response

Dear Editor,

We thank you and the reviewers for the helpful comments and critiques. We have revised the manuscript in response to the comments. The changes are highlighted in blue in the revised manuscript. We have also prepared a point-by-point reply to all the issues that were raised by the reviewers.

Please do not hesitate to contact us should you require further information.

Sincerely,

Romina Nassini

Reviewer 2:

Marini et al submitted a review entitled « TRP channels in cancer: signalling mechanisms and translational approaches ».

Major points:

Although this review is well documented, it is presented as a catalog of the role of TRP channels in cancer. This review fails to deliver on the state of research into the use of TRP channel modulators in oncology as suggested in the title and abstract.

To improve the interest of the review, the authors have to:

-          add summary diagram(s) of TRP-activated transduction mechanisms involved in cancer cell proliferation, migration or drug response.

We thank the reviewer for the suggestion. New information regarding TRP transduction mechanisms has been added in the new version of the table (Pages 13).

-          Describe and name TRP modulators and report clinical trials in progress and those with known results.

An updated section regarding clinical trials has been added (Page 18 Lines 624).

A discussion concerning the involvement of multiple TRP channels in the same type of cancer (breast, prostate, lung), would be interesting to bring to the reader.

As suggested, the role of TRP channels in the same type of cancer is now addressed in the conclusions section (Page 18 Line 660).

Minor points

A schematic picture of the TRP channel subfamily structures would be appreciated.

We thank the reviewer for the suggestion. We have added a schematic picture of the TRP channel subfamily (Figure 1).

Page 4 line 184: TRPC cells instead of TRPC channels

The typo has been corrected (Page5 Line 189).

Round 2

Reviewer 1 Report

Authors have been responsive to my comments. The modifications increase the impact of the review. 

Reviewer 2 Report

The authors have correctly answered the comments